# A Severe Systemic Infection in a 14-Year-Old Boy That Took Place during the COVID-19 Pandemic

**DOI:** 10.3390/children9050726

**Published:** 2022-05-15

**Authors:** Adam Główczewski, Przemysław Gałązka, Agata Peikow, Anna Kojro-Wojcieszonek, Dominika Tunowska, Aneta Krogulska

**Affiliations:** 1Department of Pediatrics, Allergology and Gastroenterology, Collegium Medicum in Bydgoszcz, Nicolaus Copernicus University, 87-100 Toruń, Poland; a.glowczewski@cm.umk.pl (A.G.); aneta.krogulska@cm.umk.pl (A.K.); 2Clinical Department of General and Oncological Surgery for Children and Adolescents, Collegium Medicum in Bydgoszcz, Nicolaus Copernicus University, 87-100 Toruń, Poland; galazkaprzemek@hotmail.com; 3Department of Paediatrics and Cardiology, Regional Children’s Hospital, 85-667 Bydgoszcz, Poland; agatapeikow@gmail.com (A.P.); aniakw@upcpoczta.pl (A.K.-W.)

**Keywords:** systemic infection, self-isolation, endocarditis, meningitis, muscle abscess

## Abstract

Introduction: Since March 2020, the COVID-19 pandemic has been a global talking point. Access to health care has become more difficult, and such an obstacle increase the risk of inadequate medical care, especially among paediatric patients. Case: This report describes the case of a previously healthy teenager who was staying home for 2 months due to a strict lockdown period in the COVID-19 pandemic and was admitted to hospital for fever, nausea and lumbar pain. He was diagnosed consecutively with meningitis, sepsis, paraspinal abscesses and endocarditis. Further investigation did not reveal any risk factors or immunodeficiency in the patient. Discussion: Sepsis is defined as the presence of systemic inflammatory response syndrome (SIRS) associated with a probable or documented infection. It is the leading cause of death from infection, especially if not recognized and treated quickly. Sepsis may lead to various complications, such as infective endocarditis, meningitis and abscesses. Although such complications may initially be latent, they can promote internal organ dysfunction and the possibility of their presence should be considered in any patient with systemic infection. Any child with a fever should be treated as one with the possibility of developing a severe infection. Conclusion: The presented case shows that even a previously healthy child staying in long-term home isolation can develop a severe infection with multiorgan complications, and the COVID-19 pandemic should not extend the diagnostic process in patients with symptoms of infection.

## 1. Introduction

Since March 2020, the COVID-19 pandemic has been a global talking point. Despite fever and upper respiratory tract infection symptoms, the course of COVID-19 in children is usually mild and does not lead to any long-term complications. During the pandemic, access to health care has become more difficult, and such an obstacle increases the risk of inadequate medical care, especially among paediatric patients. However, it should be emphasized that children still suffer from diseases other than SARS-CoV-2 infection, and although most are mild respiratory, digestive and urinary tract infections, children are also at risk of systemic infections, including those in the bloodstream.

It is possible that live bacteria can be present in the blood, a condition known as bacteraemia. While this can often proceed without clinical symptoms, it can also result in sepsis. Such cases require quick diagnosis and immediate implementation of appropriate treatment. The main symptoms include fever, tachycardia or bradycardia, respiratory failure and disturbed consciousness [1]. Due to the non-specific nature of these symptoms, quick diagnosis can be a challenge for primary care physicians and for paediatricians working in emergency rooms. Bleeker et al. report that sepsis was diagnosed in 8% of children treated in the paediatric intensive care unit, and that the source of infection and type of pathogen was unclear in one third of these patients [2].

This report describes a case of a previously healthy teenager with severe sepsis and multiorgan complications who was admitted to hospital during a strict lockdown period during the COVID-19 pandemic.

## 2. Case Presentation

A 14-year-old boy, so far healthy, was admitted to hospital in May 2020 for fever, nausea, and pain in the lumbar region. These symptoms had manifested three days before admission and were accompanied by deterioration of general condition on the day of admission, with concomitant disturbances of consciousness.

During the previous two months, the boy had been isolated at home with his healthy parents and grandfather due to the SARS-CoV-2 pandemic. He had no contact with his peers. However, two months earlier, he had undergone root canal treatment.

At the time of admission to the hospital, the boy was stuporous, in a serious condition. Physical examination revealed pale skin, petechiae mainly on the lower limbs and in the lumbosacral region, dry oral mucosa, reddened throat and conjunctiva, tachycardia 150 beats/minute, positive meningeal signs and a fever of 39 °C. A polymerase chain reaction (PCR) test for SARS-CoV-2 infection was negative.

The laboratory test results are given in Table 1. *Staphylococcus aureus* was successfully cultured from the blood and cerebrospinal fluid.

Sepsis and meningitis of *S. aureus* aetiology were diagnosed. Intravenous ceftriaxone and vancomycin were introduced. On the first day of hospitalization, the boy’s general condition improved, and the fever and meningeal signs resolved; however, the pain in the lumbosacral region radiating to the left lower limb persisted. On the sixth day of hospitalization, fever had risen to 38.5 °C. Laboratory tests were normalized. Magnetic resonance imaging (MRI) of the lumbosacral spine revealed the presence of skeletal muscle abscesses (Figure 1). The antibiotic therapy was changed to cloxacillin and trimethoprim with sulfamethoxazole. The abscesses were drained. *S. aureus* and *Pseudomonas aeruginosa* were cultured from a sample of the operative material, and ceftazidime was added to the treatment. Low-grade fever persisted throughout the postoperative period.

On day 3 after surgery, physical examination revealed a loud systolic murmur above the heart. Based on the clinical course and preliminary echocardiography, infective endocarditis (IE) was suspected. Transthoracic echocardiography revealed an abscess and mitral valve regurgitation (Figure 2). Laboratory tests showed an elevated concentration of NT-pro BNP (519 pg/mL). Spironolactone was administered, and antibiotic therapy was continued for a total of 40 days. The boy’s general condition improved, and normalization of laboratory parameters was achieved. After four weeks of treatment, the abscess was no longer visible, as confirmed by transesophageal echocardiography.

The patient was discharged to home in good general condition. During the follow-up a slight mitral regurgitation, but no abscesses in the area of the spine or pelvis in MRI, were found.

Diagnosis for immune disorders revealed no primary or secondary immunodeficiency. During the one-year follow-up period, the boy remained in good general condition, without any disturbing symptoms.

## 3. Discussion

Sepsis is defined as life-threatening organ dysfunction caused by a dysregulated host response to infection [1]. It is the leading cause of death from infection, especially if not recognized and treated quickly. Mortality varies from 1–20% depending on the severity of the course, the complications, and the age of the patient. Sepsis may lead to various complications, such as infective endocarditis, meningitis and abscesses. Although such complications may initially be latent, they can promote internal organ dysfunction and the possibility of their presence should be considered in any patient with systemic infection [2].

Our patient was diagnosed with sepsis and meningitis on day 1 of hospitalization.

Meningitis is an acute infection of the subarachnoid space, involving the meninges, and is associated with high mortality. The most common symptoms of meningitis are high fever, headache, vomiting, irritability, hyperaesthesia, photophobia, confusion and muscle pain [3].

The pain in the lumbar region reported by our patient was initially associated only with meningitis; however, despite treatment and a decrease in inflammatory parameters, its persistence remained a cause for concern and required further diagnostics. MRI identified abscesses in the lumbar region. This rare condition can be primary or secondary. Primary abscesses arise as a result of the spread of pathogens from distant locations, e.g., infectious endocarditis, sinusitis or myositis, via the blood or lymphatic route, while secondary abscesses typically arise as a result of direct trauma or excessive exercise. It seems that the abscesses observed in our patient were of the former type, as an interview showed that the boy had not suffered any trauma and did not play sports [4].

The appearance of a heart murmur on day 10 of treatment indicated the development of IE. The development of IE is favoured by the presence of comorbidities including heart defects, implantation of artificial cardiosurgical materials or immunodeficiency [5]. Currently, approximately 8–10% of the IE cases that develop in the paediatric population are not associated with structural heart disease or other tangible risk factors. In most cases, the infection demonstrates an *S. aureus* aetiology, as in the presented case [6]. The course and clinical symptoms of the disease may be rapid, subacute or chronic, with nonspecific symptoms and persistent fever. Musculoskeletal symptoms are common in the course of IE. Muscle pain is reported in 12–15% of patients, back pain in 13%, and joint pain in 10% [7].

The present case was diagnosed with meningitis, sepsis, paravertebral abscesses and IE; consequently, it was difficult to establish the root cause. However, it is likely that that the starting point was sepsis, followed by meningitis, skeletal muscle abscesses, and ultimately IE.

Regarding the source of sepsis, it is again most likely that infection occurred during root canal treatment, as the patient was in strict home isolation for two months due to the COVID-19 pandemic. It has been proposed that bacteraemia may be brought on by dental procedures, both those performed in dental surgeries and those taking place as part of daily oral hygiene. This is due to the specificity of the plaque biofilm, which can grow into an increasingly pathogenic bacterial flora resulting in gingivitis. In such cases, even minimal mechanical damage to this area can lead to the development of bacteraemia. One study estimated that brushing teeth twice a day for a year is associated with a several times greater risk of bacteraemia than an individual tooth extraction [8]. It should be emphasized that the potential for serious blood-borne infections, such as meningitis or IE, cannot be ruled out in any child, even healthy children, without chronic diseases.

Any child with a fever should be managed as one with the possibility of developing a severe infection. In case of doubt as to the cause of the fever, it is advisable to conduct a quick diagnosis. Despite the ongoing pandemic, a diagnosis of SARS-CoV-2 infection should not be allowed to extend this process. Even a previously healthy child in home isolation can develop a severe infection leading to dangerous complications, as in the case of the present patient.

The authors have stated explicitly that there are no conflicts of interest in connection with this article.

## Figures and Tables

**Figure 1 children-09-00726-f001:**
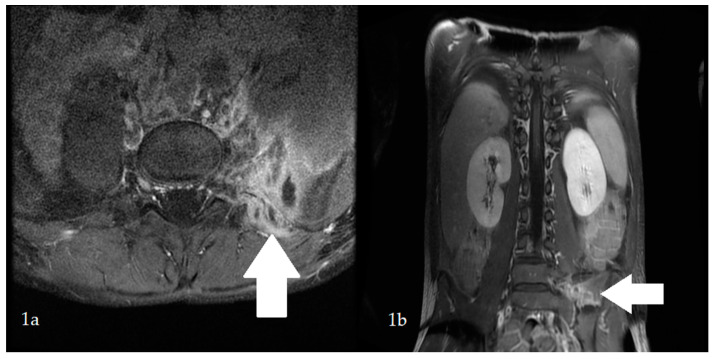
Magnetic resonance imaging (MRI) of the lumbosacral spine in the transverse (**1a**) and longitudinal (**1b**) position. Abscesses (marked with white arrows) of 10 × 18 mm occupying an area of 37 mm × 39 mm × 61 mm can be seen within the left iliopsoas muscle and paraspinal muscles at the L5-S1 level.

**Figure 2 children-09-00726-f002:**
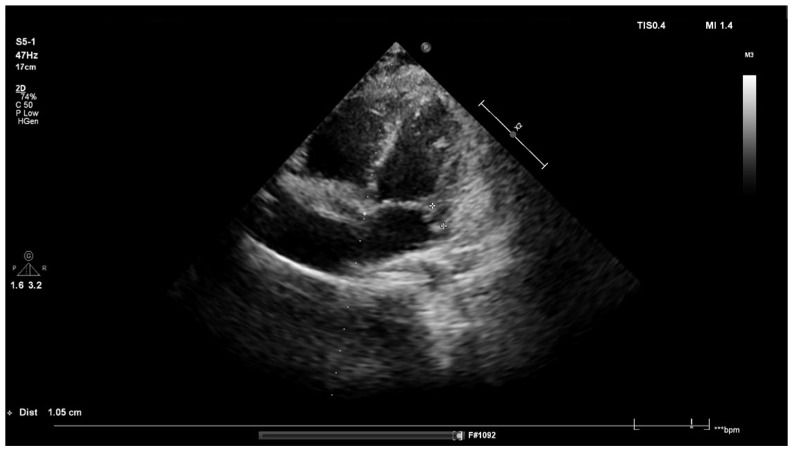
Echocardiographic examination: a hypoechoic spherical space of (cross-shaped markers) 9–10 mm in diameter can be seen at the root of the posterior leaflet of the mitral valve, which may correspond with an abscess. Moderate mitral regurgitation 8–9 mm can also be observed, reaching the apex of the left atrium.

**Table 1 children-09-00726-t001:** Results of laboratory tests on the day of admission to the hospital and at day 6.

Source	Parameter	Standard	Day 1	Day 6
Peripheral blood	WBC (×10^3^/µL)	4.5–13.5	1.85	11.63
	PLT (×10^3^/µL)	175–345	91	294
	HGB (g/dL)	12–15	10.7	14
	CRP (mg/L)	<5	236	30.38
	PCT (ng/L)	<0.5	21.42	0.23
Cerebrospinal fluid	Appearance	colorless	opalescent	colorless
	WBC (cells/µL)	<10	902	8
	Protein (mg/dL)	15–45	88.2	31.8
	Chloride (mmol/L)	115–130	121.7	118.6
	Glucose (mg/dL)	50–80	69.6	55.6
	Erythrocytes (cells/µL)	0	5	18
	Free hemoglobin	negative	negative	negative
	Smear	-	segmented neutrophils 86%, monocytoid cells 12%, banded neutrophils 2%	lymphocytes, segmented neutrophils, monocytoid cells, macrophages *

WBC—white blood cells, PLT—thrombocytes, HGB—hemoglobin, CRP—C-Reactive Protein, PCT—procalcitonin; * percentages not established due to low cytosis.

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
