# Peer review of "A Severe Systemic Infection in a 14-Year-Old Boy That Took Place during the COVID-19 Pandemic"

_children, 2022, doi:10.3390/children9050726_

Round 1

Reviewer 1 Report

The authors have described an interesting case that invites reflection on inadequate medical care following the COVID-19 emergency, which has shifted most medical attention to SARS-COV-2 infection, sometimes neglecting important clinical pictures. The manuscript is well written, fluent, and clear in all its parts.

Author Response

We would like to thank you for your support and favorable opinion on our case report. Our final hope is that this manuscript will be accepted for publication in Children, which would be considered a great honor to us. Thank you for your time and consideration.

Reviewer 2 Report

In this paper the authors present a case report of a 14-year old boy diagnosed with with meningitis, sepsis, paraspinal abscesses, and endocarditis during the pandemic of COVID-19. The case report might be interesting for the readers, however, several corrections could make a clearer message of the manuscript.

Firstly, several language corrections are required as some phrases and sentences are difficult to be understood?

Major issues:

  1. The authors based the sepsis diagnosis on SIRS, which is no longer used (please analyse the Surviving Sepsis Campaign with a new definition of sepsis).
  2.  Line 30-31: SARS-30 CoV-2 infection is often severe and can cause long-term complications in adults, as well as in children. This sentence is confusing. In children the course of COVID-19 is usually mild. Please edit.
  3. Case presentation: when was the boy admitted? Please provide the date, as the pandemic has already been lasting for over 2 years.
  4. Table 1. According to the caption, the results are presented at the discharge. In the Table, there are data from day 6, and the boy has been discharged after 40 days. Please correct.
  5. Please provide more data on the laboratory testing of the CSF, including the CSF smear. In addition, the normal range of WBC is usually  < 5 cells. Please discuss.
  6. Line 95: Diagnosis for immune disorders revealed no abnormalities. Which testing were performed, and which diseases were excluded? Did you performed testing towards HIV infection or only primary immunodeficiencies?
  7. The information on the one-year follow-up is interesting and of high value.
  8. Conclusion: Any child with a fever, should be treated as one with the possibility of developing a severe infection. It is confusing in this form. It suggests, we should TREAT every child with fever as it had sepsis (wide spectrum antibiotics, etc). Most cases of fever in children result from viral infection. Please rewrite. Maybe replacing TREATMENT with MANAGEMENT would be useful.
  9. No data on the testing towards SARS-CoV-2 was given. Please provide.
  10. Did the authors considered other possible sources of the infection? E.g, IDU?

Author Response

We would like to thank you for all the valuable comments you have provided. Basing on them we have modified the manuscript. Now we would like to reply to all your comments:

Point 1: The authors based the sepsis diagnosis on SIRS, which is no longer used (please analyse the Surviving Sepsis Campaign with a new definition of sepsis).

Response 1: We have updated the definition of sepsis basing on Surviving Sepsis Campaign.

Point 2: Line 30-31: SARS-30 CoV-2 infection is often severe and can cause long-term complications in adults, as well as in children. This sentence is confusing. In children the course of COVID-19 is usually mild. Please edit.

Response 2: We have edited the sentences according to your suggestion.

Point 3: Case presentation: when was the boy admitted? Please provide the date, as the pandemic has already been lasting for over 2 years.

Response 3: We have added relevant information in the case presentation.

Point 4: Table 1. According to the caption, the results are presented at the discharge. In the Table, there are data from day 6, and the boy has been discharged after 40 days. Please correct.

Response 4: We have corrected the caption.

Point 5: Please provide more data on the laboratory testing of the CSF, including the CSF smear. In addition, the normal range of WBC is usually  < 5 cells. Please discuss.

Response 5: We have provided information about other CSF parameters, including eryhtrocytes, free hemoglobin, concentration of glucose, chlorides, and the CSF smear. In the case of the CSF analyse at day 6 the method used by laboratory did not allow to provide the smear result in percentages due to low cystosis.

The normal range of WBC provided by our hospital laboratory is 0-10/ul. The values of <30 and 23 were typographical errors and we have corrected them.

Point 6: Line 95: Diagnosis for immune disorders revealed no abnormalities. Which testing were performed, and which diseases were excluded? Did you performed testing towards HIV infection or only primary immunodeficiencies?

Response 6: During the hospitalisation in our Department the the basic diagnostic towards immunodeficiencies (major classes and subcasses of immunoglobulins, lymphocites subpopulations, HIV) was provided. During follow-up the boy was hospitalised in Hematology and Oncology Department for more detailed immunological diagnostics, including NGS testing for immunodeficiencies, which did not reveal any abnormalities.

We have added “primary and secondary” immunodeficiencies to the sentence.

Point 7: The information on the one-year follow-up is interesting and of high value.

Response 7: We fully agree.

Point 8: Conclusion: Any child with a fever, should be treated as one with the possibility of developing a severe infection. It is confusing in this form. It suggests, we should TREAT every child with fever as it had sepsis (wide spectrum antibiotics, etc). Most cases of fever in children result from viral infection. Please rewrite. Maybe replacing TREATMENT with MANAGEMENT would be useful.

Response 8: We have ammended the sentence according to your accurate suggestion.

Point 9: No data on the testing towards SARS-CoV-2 was given. Please provide.

Response 9: We have added the information. The boy was tested for SARS-CoV-2 at the day of admission.

Point 10: Did the authors considered other possible sources of the infection? E.g, IDU?

Response 10: The detailed history taken from parents and the patient along with accurate physical examination did not indicate any other possible source of the infection, including IDU.

We would like thank you once again for all your help and support. We truly believe your comments have helped to make this article more valuable. Our final hope is that this manuscript will be accepted for publication in Children, which would be considered a great honor to us. Thank you for your time and consideration.

Round 2

Reviewer 2 Report

I would like to thank the authors for all the corrections. The manuscript conveys now a much clearer message. Still some language corrections would be helpfull. I have no further queries.